Subject Areas:
psychology

Keywords:
grammatical decision task, sentence priming, transposed-word priming

Author for correspondence:
Jonathan Mirault
e-mail: jonathan.mirault@univ-amu.fr

# Fast priming of grammatical decisions: repetition and transposed-word priming effects

Jonathan Mirault[1], Mathieu Declerck[2] and Jonathan Grainger[1]

[1]Laboratoire de Psychologie Cognitive, Aix-Marseille Université and Centre National de la Recherche Scientifique, Marseilles, France
[2]Linguistics and Literary Studies, Vrije Universiteit Brussel, Brussels, Belgium

(iD) JM, 0000-0003-1327-7861

We used the grammatical decision task to investigate fast priming of written sentence processing. Targets were sequences of 5 words that either formed a grammatically correct sentence or were ungrammatical. Primes were sequences of 5 words and could be the same word sequence as targets, a different sequence of words with a similar syntactic structure, the same sequence with two inner words transposed or the same sequence with two inner words substituted by different words. Prime-word sequences were presented in a larger font size than targets for 200 ms and followed by the target sequence after a 100 ms delay. We found robust repetition priming in grammatical decisions, with same sequence primes leading to faster responses compared with prime sequences containing different words. We also found transposed-word priming effects, with faster responses following a transposed-word prime compared with substituted-word primes. We conclude that fast primed grammatical decisions might offer investigations of written sentence processing what fast primed lexical decisions have offered studies of visual word recognition.

## 1. Introduction

The field of visual word recognition has benefitted from priming methodology for more than half a century. Beginning with the seminal work of Meyer & Schvaneveldt [1], priming research has dominated this field, with multitudes of articles investigating semantic, orthographic, phonological and morphological priming with single words. Another seminal priming study, particularly relevant for the present work, is that of Neely [2] that provided the crucial demonstration of how prime duration impacts the relative

contribution of automatic versus controlled (strategic) processes [3,4] in driving semantic priming effects. Neely's study pointed to a 250 ms stimulus onset asynchrony (SOA) as a critical point above which more controlled processing could partly determine priming effects. Neely found that relative to a neutral prime condition, inhibitory effects of semantically unrelated primes disappeared at that SOA while facilitatory effects of related primes remained. The controlled processes operating at longer SOAs could involve several strategies, such as trying to guess target word identity on the basis of prime word identity, or biasing responses when participants notice the relation between prime and target stimuli (e.g. in a lexical decision task, if a semantic relation is detected, then the target must be a word). In general, these are thought to be slow inferential processes, compared with the rapid, automatic processing thought to underly the linguistic processes of interest. Although much shorter prime durations (of the order of 50–60 ms) were later to become the gold standard in word recognition research [5], Neely's demonstration is important for studies, such as the present one, where extremely short prime exposure durations are inconceivable.

The starting point of the present work is the observation that investigations of sentence-level comprehension have not adopted priming methodology to the same extent as single-word recognition research. The most obvious reason for this state of affairs is that sentence processing takes too long for entire sentences to be used as prime stimuli, given that we know from single-word studies that presenting primes for too long a duration leads to strategic influences on the observed results [2]. Nevertheless, priming methodology has been used to investigate syntactic processing with written sentences within the field of research known as syntactic or structural priming (e.g. [6–8]). The results of these studies indicate that residual activation of the syntactic structure induced during prime sentence processing can impact subsequent target sentence processing. However, in all these studies, prime sentences were 4-words long or longer and participants had to read the sentences, which would take at least 1 s. This would leave ample room for contamination of the observed priming effects by controlled processes. Here, we aim to demonstrate that much shorter prime durations might be an interesting option in studies of written sentence processing.

There are two key findings that motivated the design of the current research. The first is that recent research has revealed that sentence-level information can be extracted from very briefly presented written sentences [9–11]. In these studies, 4-word sequences were presented simultaneously to participants for 200 ms and followed by a backward mask. The sequences could either be grammatically correct (e.g. the boy can run) or an ungrammatical sequence composed of the same words (e.g. run boy can the), and participants had to identify one word at a post-cued location (e.g. the word 'boy' at position 2 in these examples). Target word identification accuracy was found to be significantly greater when the sequence was grammatically correct. These findings suggest that word identification can proceed in parallel and that word identities are rapidly associated with their corresponding parts of speech. In turn, this should enable the fast computation of an approximate, 'good-enough', syntactic structure (when available) that then provides feedback to on-going word identification processes within a cascaded, interactive system for sentence processing [11].

The second key finding is that recent research has revealed the interest of a new task applied to sentence-level processing that can be thought of as the equivalent of the lexical decision task for word-level processing. This is the grammatical decision task. This task was first put to use by Mirault et al. [12] in order to demonstrate transposed-word effects during written sentence processing. Mirault et al. found that decisions to ungrammatical word sequences were harder to make when the sequence was created by transposing two adjacent words in a grammatical sequence (e.g. the white *was cat* big) compared with similar ungrammatical sequences for which a correct sentence cannot be formed by transposing any two words (e.g. the white was cat slowly). In the standard response-limited version of the grammatical decision task, word sequences remained on screen until participants' response, hence mimicking the standard version of the lexical decision task used to study single-word reading. However, and highly relevant with respect to the present work, Mirault & Grainger [13] used a data-limited version of the grammatical decision task in order to investigate how accuracy in making such decisions varied as a function of stimulus presentation time. Targets were randomly intermixed grammatical and ungrammatical 5-word sequences and presentation time varied randomly between 50 ms and 500 ms with 50 ms steps. Mirault and Grainger found that accuracy in grammatical decisions was practically at asymptote with a presentation duration of 300 ms. At that duration, accuracy to grammatical sequences was on average 87% with a d' of 2.

In the present study, we therefore hypothesized that presenting word sequences for 200 ms (with a total SOA of 300 ms due to a 100 ms blank screen between primes and targets) should prime grammatical decisions to subsequent target sequences, much like priming effects on single-word recognition obtained with the lexical decision task. We further reasoned that with a 300 ms SOA, we

were well within the limits of automatic processing for 5-word sequences, compared with the estimated 250 ms SOA for single-word presentations. We decided to first test this new fast priming paradigm applied to sentence-level processing with two conditions that were the most likely to reveal priming effects: repetition priming and transposed-word priming. Indeed, at the level of single-word processing, the strongest priming effects obtained with very brief prime durations are repetition priming (e.g. [5,14]) and transposed-letter priming (e.g. [15]; see Grainger [16] for a review).[1]

# 2. Material and methods

## 2.1. Participants

Ninety-six[2] participants (female = 41) were recruited online on the Prolific platform [18]. They ranged in age from 18 to 72 years ($M = 31.76$, s.d. = 12.21) and were naive to the purpose of the experiment. They received monetary compensation (£7.50/h) and signed an informed consent form in accordance with the provisions of the World Medical Association Declaration of Helsinki prior to the experiment. Ethics approval was obtained from the Comité de Protection des Personnes SUD-EST IV (no. 17/051).

## 2.2. Stimuli and design

We first selected 360 5-word sentences in French. They ranged in length between 17 and 45 characters (including spaces; $M = 28.73$; s.d. = 4.12). The design involved two independent manipulations of prime-target relatedness—one for repetition priming and one for transposed-word priming. In the repetition priming conditions, the related prime was identical to the target and the unrelated primes were formed by changing the words in the target sequence with other words of similar length and syntactic category. In the transposed-word priming condition, related primes were formed by transposing two inner words in the target, and the unrelated primes were formed by replacing the same two words with different words matched in length. We also created 360 ungrammatical sequences of 5 French words that were included for the purpose of the grammatical decision task. The primes associated with the ungrammatical targets mimicked the four types of primes used for the grammatical targets. Table 1 provides examples of the different prime conditions for grammatically correct and ungrammatical target sequences (the complete list of the stimuli are available online on the OSF site). Data for the grammatical targets were analysed separately from the ungrammatical targets, as is typical of analyses of lexical decision data and grammatical decision data. The design involved two separate analyses of prime-target relatedness, one for the repetition condition and one for the transposition condition. Contrary to what was stated in the pre-registration form, we did not opt for a $2 \times 2$ design, given the difference in the degree of prime-target overlap in the unrelated repetition condition (0 word overlap) and the unrelated transposition condition (3/5 words overlap).

## 2.3. Apparatus and procedure

The experiment was created using the online experiment builder LabVanced [19] and displayed on participants' computer screens (we allowed Mac, Windows and Linux operating systems, but tablets and smartphones were disabled). Before the start of the experiment, a consent form was displayed on the screen to inform participants that their data would be recorded anonymously. If they accepted to participate, the instructions were presented on the screen and were followed by 10 practice trials. The main experiment was composed by 720 trials, with a different random order for each participant. Each trial started with a fixation cross in the middle of the screen for a duration of 250 ms. Following a 200 ms blank screen, the prime sequence was rapidly presented (200 ms) followed by another blank screen for 100 ms (i.e. a 300 ms SOA). The target sequence remained on the screen until the participant indicated whether the target sequence was grammatical or ungrammatical by pressing the

---

[1]The hypotheses and methodology were pre-registered on the Open Science Framework (OSF) repository (https://osf.io/ba4nw).

[2]1000 simulations were run based on a pilot study of 16 participants to determine the necessary number of participants. The results of the power analysis showed that with 69 participants, power was 80% to observe the main effects on which the current study focuses. We decided to collect data from 80 participants (see pre-registration form) to cover for any issues with online experiments and in order to increase power above 80% as suggested by Brysbaert [17]. Some more participants were tested than the pre-registered 80 because we were expecting a number of participants to be dropped due to, for instance, poor accuracy or not finishing the experiment. However, far fewer participants had to be excluded based on these criteria. Since we had no ground to exclude these additional participants, we kept them in the analyses. The actual power values are provided in the Results section.

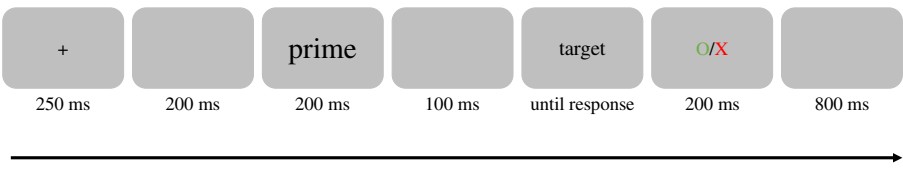

**Figure 1.** Procedure of one experimental trial. Primes and targets were sequences of five words presented simultaneously and that could be grammatically correct or not. Participants made speeded grammatical decisions to the target sequence.

**Table 1.** Examples of prime stimuli for the four priming conditions for the grammatically correct target 'Antoine ne sait pas tout' and the ungrammatical target 'Antoine ne sait oeil tout'.

| | type of relation | |
|---|---|---|
| relatedness | repetition | transposition |
| grammatical targets | | |
| related | Antoine ne sait pas tout | Antoine sait ne pas tout |
| unrelated | Adeline se voit plus loin | Antoine vaut ça pas tout |
| ungrammatical targets | | |
| related | Antoine ne sait oeil tout | Antoine sait ne oeil tout |
| unrelated | Camille ne voit café près | Antoine mime ni oeil tout |

arrows on their keyboard (right for correct, left for incorrect). There was no time limit, and each trial required a response. Finally, feedback was presented to inform the participant of their performance (a green circle for correct responses and a red cross for incorrect responses) for 200 ms followed by a blank screen for 800 ms. Prime sequences were presented in a larger font size (20-point) than target sequences (16-point) in order to avoid purely visual overlap in the related conditions. A visual depiction of a sample trial is shown in figure 1. A pause screen was displayed after every 60 trials and participants had to click any key to continue the experiment.

## 2.4. Analysis

We used linear mixed-effects (LMEs) models to analyse response times (RTs), with items and participants as crossed random effects (including by-item and by-participant random intercepts) [20] and with random slopes [21]. Generalized (logistic) linear mixed-effects (GLMEs) models were used to analyse error rates. The separate models for the repetition and transposition factors[3] were fitted with the lmer (for LMEs) and glmer (for GLMEs) functions from the lme4 package [22] in the R statistical computing environment. We report regression coefficients ($b$), standard errors (s.e.) and $t$-values (for LMEs) or $z$-values (for GLMEs) for the Relatedness factor (i.e. priming effects). Fixed effects were deemed reliable if $|t|$ or $|z| > 1.96$ [23], and the corresponding values are presented in bold font. RTs were log10-transformed prior the analysis since RTs did not follow a normal distribution according to Kolmogorov–Smirnov normality test.

## 3. Results

We analysed RTs (in ms) for correct responses, which is the time between the onset of target presentation until participants' response, and error rate (in percentage). Prior to analyses, we excluded four participants with an average error rate greater than 25%. We also removed trials with RT values less than 100 ms or greater than 5000 ms (2.06% of trials). Separate analyses were performed for Repetition

---

[3]Contrary to what was indicated in the pre-registration form, we decided to perform separate analyses for the repetition and transposed-word priming effects because the two related conditions are inherently different. The same pattern of significance was observed when using a 2 (related versus unrelated) × 2 (repetition versus transposition) analysis, which was in the pre-registration form.

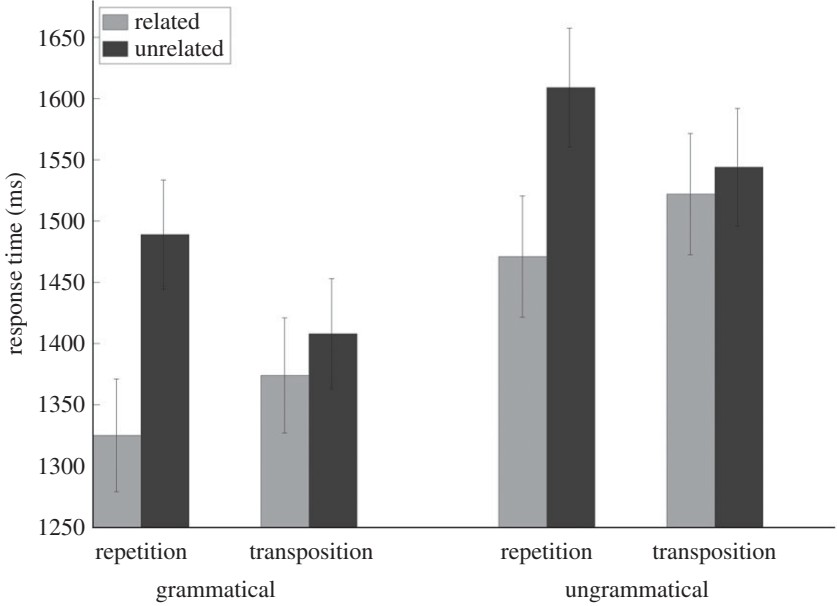

**Figure 2.** Mean RT per experimental condition. Error bars are within-participant 95% confidence intervals [24].

priming and Transposed-word priming. The two models, for the Repetition and Transposed-word priming effects, reach, respectively, a statistical power of 100% (IC = 0.37) and 99.20% (IC = 1.22).

## 3.1. Response time

Prior to the RT analysis, we removed trials with incorrect responses (5.30%) and trials with RTs beyond 2.5 s.d. from the grand mean (3.38%). The remaining dataset was composed of 29 600 observations. Condition means are reported in figure 2.

For grammatically correct targets, there was a significant effect of Relatedness in the repetition condition ($b = 0.13$; s.e. = 0.006; $t = $ **20.64**), with faster RTs to targets following related primes ($M = 1456$ ms; 95% CI = 19.06) compared to unrelated primes ($M = 1588$ ms; 95% CI = 20.15). There was also a significant effect of Relatedness in the transposition condition ($b = 0.02$; s.e. = 0.006; $t = $ **4.13**), with faster RTs to targets following related primes ($M = 1505$ ms; 95% CI = 20.89) compared to unrelated primes ($M = 1524$ ms; 95% CI = 18.91).

For the ungrammatical target sequences, there was a significant effect of relatedness in the repetition condition ($b = 0.10$; s.e. = 0.008; $t = $ **12.62**), with faster RTs to targets following related primes ($M = 1309$ ms; 95% CI = 19.81) compared to unrelated primes ($M = 1468$ ms; 95% CI = 16.01). There was also a significant effect of relatedness in the transposition condition ($b = 0.02$; s.e. = 0.008; $t = $ **2.54**), with faster RTs to targets following related primes ($M = 1357$ ms; 95% CI = 18.68) compared to unrelated primes ($M = 1390$ ms; 95% CI = 18.18).

## 3.2. Error rates

The dataset for the error analysis consisted of 32 352 observations. Average values per conditions are reported in table 2. For the grammatically correct targets, there was a significant effect of Relatedness in the repetition condition ($b = 0.28$; s.e. = 0.11; $z = $ **2.42**) with more errors in the unrelated condition ($M = 4.96$; 95% CI = 0.58) compared to the related condition ($M = 3.92\%$; 95% CI = 0.79). The effect of Relatedness was not significant in the transposition condition ($b = 0.09$; s.e. = 0.11; $z = 0.84$).

For the ungrammatical targets, the effect of Relatedness was not significant for either the repetition condition ($b = 0.08$; s.e. = 0.13; $z = 0.60$) or the transposition condition ($b = 0.04$; s.e. = 0.10; $z = 0.48$).

## 3.3. Distributional analysis of priming effects

We further analysed the distribution of the repetition and transposition priming effects in RTs as a function of the overall distribution of RTs (delta plots; e.g. [25]). The results are shown in figure 3.

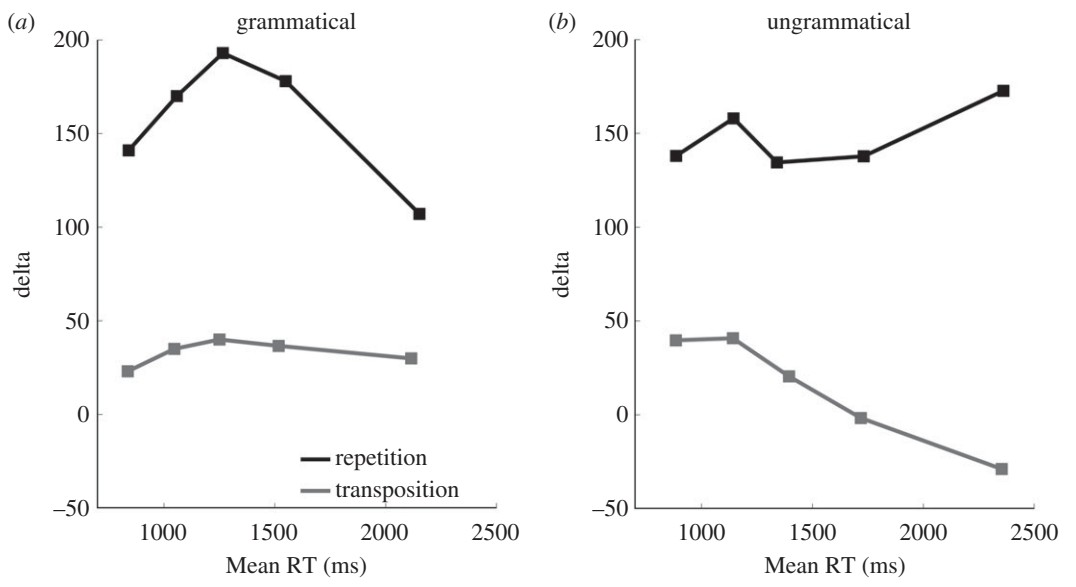

**Figure 3.** Distribution of priming effects (delta plots) for the four types of priming examined in the present study. These plots show how the size of priming effects (RT difference between related and unrelated primes) varies as a function of the overall distribution of RTs in a given condition, for the 0.1, 0.3, 0.5, 0.7 and 0.9 quantiles.

**Table 2.** Error rates (percentages) per experimental condition. Note that values in parentheses are within-participant 95% confidence intervals [24].

|  | grammatical | | | ungrammatical | | |
|---|---|---|---|---|---|---|
|  | related | unrelated | effect | related | unrelated | effect |
| transposition | 4.60 (0.81) | 4.19 (0.68) | −0.41 | 6.34 (0.90) | 5.93 (0.73) | −0.41 |
| repetition | 3.92 (0.79) | 4.96 (0.58) | 1.04 | 6.30 (0.83) | 6.16 (1.06) | −0.14 |

This analysis revealed that the repetition priming effects were quite substantial across the entire RT distribution, for both grammatical and ungrammatical targets. For grammatical targets, the transposition priming effects were again stable across the entire RT distribution, but for ungrammatical targets these priming effects disappeared with longer RTs.

### 3.4. *Post hoc* analysis

Given the design of the present experiment, we were able to compare performance in the related repetition and the related transposition conditions for both grammatical and ungrammatical targets (table 1). We found a significant difference in RTs to the grammatically correct targets ($b = 0.04$; s.e. $= 0.006$; $t = \mathbf{6.35}$) with faster RTs in the related repetition condition ($M = 1309$ ms; 95% CI $= 19.81$) compared to the related transposed condition ($M = 1357$; 95% CI $= 18.68$). The same effect was seen in RTs to ungrammatical targets ($b = 0.03$; s.e. $= 0.007$; $t = \mathbf{5.18}$), with again faster RTs in the related repetition condition ($M = 1456$ ms; 95% CI $= 19.06$) compared to the related transposition condition ($M = 1505$ ms; 95% CI $= 20.89$). The effects in the error rates were not significant, neither for the grammatical targets ($b = 0.13$; s.e. $= 0.14$; $z = 0.95$), nor for the ungrammatical targets ($b = 0.01$; s.e. $= 0.12$; $z = 0.12$).

## 4. Discussion

In the present study, we tested a new methodology for the study of written sentence processing. For the first time, we combined the novel grammatical decision task [12,13] with a fast priming manipulation. Target word sequences were either grammatically correct or not, and prime-word sequences could be the same as targets and were compared with a syntactically similar sequence of different words (repetition priming) or could be formed by transposing two adjacent inner words of targets and were compared with

sequences formed by substituting the same two words with different words (transposed-word priming). Crucially, prime-word sequences were presented for only 200 ms (followed by a 100 ms blank screen prior to target presentation), thus making it highly unlikely that our participants were using slow inferential processes in an attempt to improve their responses to targets. We found significant repetition and transposed-word priming effects in RTs to both grammatical and ungrammatical targets. The fact that we found the same pattern in responses to both types of targets is important because it demonstrates that our participants were not anticipating responses to targets upon detecting a relation between the prime and target sequence (e.g. I detect a relation therefore the target is grammatical). If that had been the case, then we should have seen the opposite pattern of effects across the two types of targets. The priming effects seen with ungrammatical targets suggest that 'ungrammatical' decisions are made on the basis of word identity and word order information, and that this information is one source of priming effects. Some form of whole-sentence syntactic representation must also contribute to priming effects seen in 'grammatical' decisions, and the fact that we found priming effects with a 300 ms SOA points to the very fast computation of a primitive syntactic structure (see also Mirault & Grainger [13] for converging evidence). In sum, believe that we were successful in isolating conditions that provide enough time for prime-word sequences to be processed up to sentence-level representations while limiting the contribution of strategic influences on priming effects.[4] Thus, the main contribution of the present study is the demonstration that fast priming can be used to investigate written sentence processing.

The transposed-word priming we found in the present study mimics the highly influential, and oft-replicated, transposed-letter priming effect seen in lexical decisions to single words (e.g. [16]; see Grainger [18] for a review). In the same way as transposed-letter priming effects have been taken to reflect the noisy, parallel association of letter identities to letter positions in a string of letters, we suggest that transposed-word priming reflects the noisy, parallel association of word identities to their locations in a line of text. In combination with prior reports of transposed-word effects [10,12,26,27], we take this as further evidence in favour of parallel word processing during sentence reading, or at least the proof that such parallel processing is possible [28]. We nevertheless acknowledge that taking transposed-word effects as evidence for parallel word processing is a controversial issue (see Huang & Staub [29] for a recent review). At a purely empirical level, in the present study we were able, for the first time, to demonstrate a transposed-word effect in responses to grammatically correct word sequences (all prior research has shown the effect in responses to ungrammatical target sequences).

As can be seen in figure 2, trials in the repetition condition (both related and unrelated) generated faster RTs than trials in the transposition condition. Given that the related repetition and related transposition primes were matched (table 1), we could directly compare these two conditions, and we found significant differences in RTs for both the grammatically correct targets and the ungrammatical targets. That is, although the related transposition primes facilitated target processing relative to the matched unrelated (double substitution) primes (the transposed-word priming effect), transposition primes were not as effective as repetition primes.[5] This mirrors the findings obtained with transposed-letter effects, where reading text with letter transpositions is hindered relative to normal text (e.g. [30]).

Obvious extensions of the present work include manipulations of syntactic and semantic overlap across primes and targets. Combined with a manipulation of prime duration, such research could uncover the different time-courses of syntactic and semantic processing at the sentence level. Alternatively, as has been successfully applied in the study of visual word recognition (e.g. [31,32]), combining our priming paradigm with electroencephalogram recordings could open up new horizons for studying written sentence processing. We also note that the use of the grammatical decision task might bias processing towards syntax as opposed to semantics. The use of a semantic judgement task, such as deciding whether syntactically correct word sequences make sense or not, could help discriminate between the relative contribution of syntax and semantics. Finally, in the present work, primes and targets were presented in different font sizes. In future work, it will be important to examine other kinds of purely visual changes between primes and targets, and notably a change in case. We predict that such changes should have little impact on priming effects.

---

[4]A reviewer pointed out the relatively long RTs to make grammatical decisions, which are, however, quite typical of this task (e.g. [12]. We suggest that the time that elapses between having enough information to make a reasonably accurate grammatical decision and the moment a response key is pressed is taken up by decision-level and verification processes whereby an initial fast decision is subject to subsequent checking as more information becomes available with time and eye movements.

[5]Note that only an extreme version of flexible position coding (i.e. evidence for a word at position $N$ is equivalent to the evidence for the same word at position $N + 1$) would predict no difference between repetition and transposition priming effects.

Summing up, we tested a combination of fast priming and the grammatical decision task as a novel methodology for investigating written sentence processing. We found significant repetition and transposed-word priming effects in RTs to both grammatical and ungrammatical target sequences. We therefore believe that this particular methodology might have the same impact on sentence comprehension research that the primed lexical decision task has had on visual word recognition research.

Ethics. Participants received monetary compensation (£7.50/h) and signed an informed consent form in accordance with the provisions of the World Medical Association Declaration of Helsinki prior to the experiment. Ethics approval was obtained from the Comité de Protection des Personnes SUD-EST IV (no. 17/051).

Data accessibility. Stimuli, data and analysis script are publicly available on the Open Science Framework (OSF) at: https://osf.io/ba4nw/.

Authors' contributions. J.M. was involved in design, creation of the experiment, analysis and writing. M.D. was involved in design, analysis and writing. J.G. was involved in design and writing. All authors approved the final version.

All authors gave final approval for publication and agreed to be held accountable for the work performed therein.

Competing interests. We declare we have no competing interests.

Funding. This study was funded by grant no. 742141 from the European Research Council (ERC).

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
