## [Peer Review File · Royal Society Open Science]

Review History

RSOS-211082.R0 (Original submission)

Review form: Reviewer 1

Is the manuscript scientifically sound in its present form?

Yes

Are the interpretations and conclusions justified by the results?

Yes

Is the language acceptable?

Yes

Do you have any ethical concerns with this paper?

No

Have you any concerns about statistical analyses in this paper?

No

Recommendation?

Accept with minor revision (please list in comments)

Comments to the Author(s)

Review of ms “Fast priming of grammatical decisions: Repetition and transposed-word priming effects”

This is a very straightforward manuscript presenting a novel priming task together with an experiment using the task manipulation prime-target relationships. Clearly, the findings showed a priming effect, and this priming methodology can be extended to many different scenarios, so I am very positive.

I only have a minor comment. I believe the authors can make of their data by running some more exploratory analyses to shed some light on the nature of the priming effects – which were very large for repetitions but substantially smaller for transpositions. My suggestion is to obtain the delta plots for the repetition and transposition effects, as they may offer some extra interpretive cues (e.g., an early effect in the .1 quantile; whether the effect grows in the higher quantiles for repetition effects, but much less so for transposition effects, and so on). There are many examples of the use of delta plots in priming experiments (e.g., 10.1016/j.jecp.2020.104911, to cite one instance).

--Table 1 may be better formatted (there was an isolated “l” in the .pdf)

--What are the numbers in the parentheses in Table 2?

Review form: Reviewer 2 (Kevin Paterson)

Is the manuscript scientifically sound in its present form?

Yes

Are the interpretations and conclusions justified by the results?

No

Is the language acceptable?

Yes

Do you have any ethical concerns with this paper?

No

Have you any concerns about statistical analyses in this paper?

No

Recommendation?

Major revision is needed (please make suggestions in comments)

Comments to the Author(s)

This is a very well written paper that introduces a fast priming method to investigate elements of sentence-level processing for short sentences. The study is well-designed, with good use of pre-registration methods and a transparent account of where the methods deviated from this protocol. The authors also explicitly consider statistical power and take account of this in determining sample size.

Participants have to make grammatical judgments about short word strings. These strings follow the presentation of prime strings. In one case (repetition primes), the primes are structurally

equivalent to the target and contain either the same words (the related condition) or different words (the unrelated condition). In other case (transposed-word primes), the primes include a word transposition but with either the same word (related condition) or different words (unrelated condition) to the target. The authors examine the effects of these primes on error rates and latencies for correct responses in the grammatical judgment task.

The key findings are that grammatical decisions are facilitated by the related relative to the unrelated primes in both the repetition priming and transposed-word conditions.

This is an interesting paradigm with some equally interesting results. However, there are several issues with the findings that I feel need further explanation. Also, the discussion of transposed-word effects would benefit from some discussion of alternative possibilities.

1. The key effect is that primes containing the same words facilitate grammatical decisions relative to primes with substituted words. This is particularly important in the context of the transposed-word primes as this shows that primes containing word transpositions work similarly to repetition primes, and can facilitate grammaticality decisions. This adds support to the authors claim that word order is processed flexibly in reading. I can see this is the case. However, the analyses they report do not compare the effect in repetition priming versus transposed-word priming. It isn't clear why this comparison is not made. Footnote 3 notes that this approach, which differs from the pre-registration, was because the related conditions for repetition and transposed word conditions are inherently different. I don't follow that argument, as the only difference appears to be the transposition of words to create the transposed-word primes.

Interestingly, Figure 2 appears to show longer responses for grammatical (and also ungrammatical) targets following a transposed-word prime than a repetition prime. There is also a much smaller related prime advantage for the transposed-word targets compared to the repetition targets. This seems to suggest a smaller benefit for transposed-word primes compared to repetition primes and therefore a cost for word transposition processing? How does this fit with arguments for flexible word order processing? Does it suggest this flexibility incurs a cost?

2. I'm unclear why an ungrammatical prime was used for the ungrammatical targets. Also, I'm not clear why a transposed-word ungrammatical prime should facilitate decisions for these targets relative to the substitution condition. The explanation for transposed-word effects is that flexibility in the processing of word order allows access to a grammatical representation of the input that should then facilitate the subsequent recognition of a grammatical target. However, it's unclear how a transposed-word prime should facilitate grammaticality decisions for an ungrammatical target.

3. The argument presented here is that basic information about grammatical structure of at least a portion of a sentence can be extracted quickly during reading, from a 200ms display, and that this allows a coarse or "good enough" computation of grammatical structure. However, if basic information about grammatical structure can be obtained this quickly, why are responses in the grammatical judgment task so slow, in the order of 1300 to 1500 ms for grammatical targets in the present experiment?

4. I appreciate the authors have produced a streamlined report that is focused on the main issues motivating the use of the fast priming task to study sentence-level processing. However, in discussing word transposition effects, it would be helpful to acknowledge controversies that sound this effect, including whether it relies on the parallel encoding of multiple words at a time during reading, as the authors claim, and whether the transposed word effect occurs perceptually or post-perceptually. I note, for example, the recent review article by Huang and Staub (2021).

Huang, K.-J., & Staub, A. (2021). Why do readers fail to notice word transpositions, omissions, and repetitions? A review of recent evidence and theory. *Language Linguistics Compass*, e12434. <https://doi.org/10.1111/lnc3.12434>

Decision letter (RSOS-211082.R0)

Dear Dr MIRAULT

The Editors assigned to your paper RSOS-211082 "Fast priming of grammatical decisions: Repetition and transposed-word priming effects" have now received comments from reviewers and would like you to revise the paper in accordance with the reviewer comments and any comments from the Editors. Please note this decision does not guarantee eventual acceptance.

Please submit your revised manuscript and required files (see below) no later than 21 days from today's (ie 28-Oct-2021) date. Note: the ScholarOne system will 'lock' if submission of the revision is attempted 21 or more days after the deadline. If you do not think you will be able to meet this deadline please contact the editorial office immediately.

on behalf of Professor Martin Pickering (Associate Editor) and Essi Viding (Subject Editor)
openscience@royalsociety.org

Reviewer comments to Author:

Reviewer: 1

Comments to the Author(s)

Review of ms "Fast priming of grammatical decisions: Repetition and transposed-word priming effects"

This is a very straightforward manuscript presenting a novel priming task together with an experiment using the task manipulation prime-target relationships. Clearly, the findings showed a priming effect, and this priming methodology can be extended to many different scenarios, so I am very positive.

I only have a minor comment. I believe the authors can make of their data by running some more exploratory analyses to shed some light on the nature of the priming effects – which were very large for repetitions but substantially smaller for transpositions. My suggestion is to obtain the delta plots for the repetition and transposition effects, as they may offer some extra interpretive cues (e.g., an early effect in the .1 quantile; whether the effect grows in the higher quantiles for repetition effects, but much less so for transposition effects, and so on). There are many examples of the use of delta plots in priming experiments (e.g., 10.1016/j.jecp.2020.104911, to cite one instance).

--Table 1 may be better formatted (there was an isolated "l" in the .pdf)

--What are the numbers in the parentheses in Table 2?

Reviewer: 2

Comments to the Author(s)

This is a very well written paper that introduces a fast priming method to investigate elements of sentence-level processing for short sentences. The study is well-designed, with good use of pre-registration methods and a transparent account of where the methods deviated from this protocol. The authors also explicitly consider statistical power and take account of this in determining sample size.

Participants have to make grammatical judgments about short word strings. These strings follow the presentation of prime strings. In one case (repetition primes), the primes are structurally equivalent to the target and contain either the same words (the related condition) or different words (the unrelated condition). In other case (transposed-word primes), the primes include a word transposition but with either the same word (related condition) or different words (unrelated condition) to the target. The authors examine the effects of these primes on error rates and latencies for correct responses in the grammatical judgment task.

The key findings are that grammatical decisions are facilitated by the related relative to the unrelated primes in both the repetition priming and transposed-word conditions.

This is an interesting paradigm with some equally interesting results. However, there are several issues with the findings that I feel need further explanation. Also, the discussion of transposed-word effects would benefit from some discussion of alternative possibilities.

1. The key effect is is that primes containing the same words facilitate grammatical decisions relative to primes with substituted words. This is particularly important in the context of the transposed-word primes as this shows that primes containing word transpositions work similarly to repetition primes, and can facilitate grammaticality decisions. This adds support to the authors claim that word order is processed flexibly in reading. I can see this is the case. However, the

analyses they report do not compare the effect in repetition priming versus transposed-word priming. It isn't clear why this comparison is not made. Footnote 3 notes that this approach, which differs from the pre-registration, was because the related conditions for repetition and transposed word conditions are inherently different. I don't follow that argument, as the only difference appears to be the transposition of words to create the transposed-word primes.

Interestingly, Figure 2 appears to show longer responses for grammatical (and also ungrammatical) targets following a transposed-word prime than a repetition prime. There is also a much smaller related prime advantage for the transposed-word targets compared to the repetition targets. This seems to suggest a smaller benefit for transposed-word primes compared to repetition primes and therefore a cost for word transposition processing? How does this fit with arguments for flexible word order processing? Does it suggest this flexibility incurs a cost?

2. I'm unclear why an ungrammatical prime was used for the ungrammatical targets. Also, I'm not clear why a transposed-word ungrammatical prime should facilitate decisions for these targets relative to the substitution condition. The explanation for transposed-word effects is that flexibility in the processing of word order allows access to a grammatical representation of the input that should then facilitate the subsequent recognition of a grammatical target. However, it's unclear how a transposed-word prime should facilitate grammaticality decisions for an ungrammatical target.

3. The argument presented here is that basic information about grammatical structure of at least a portion of a sentence can be extracted quickly during reading, from a 200ms display, and that this allows a coarse or "good enough" computation of grammatical structure. However, if basic information about grammatical structure can be obtained this quickly, why are responses in the grammatical judgment task so slow, in the order of 1300 to 1500 ms for grammatical targets in the present experiment?

4. I appreciate the authors have produced a streamlined report that is focused on the main issues motivating the use of the fast priming task to study sentence-level processing. However, in discussing word transposition effects, it would be helpful to acknowledge controversies that sound this effect, including whether it relies on the parallel encoding of multiple words at a time during reading, as the authors claim, and whether the transposed word effect occurs perceptually or post-perceptually. I note, for example, the recent review article by Huang and Staub (2021).

Huang, K.-J., & Staub, A. (2021). Why do readers fail to notice word transpositions, omissions, and repetitions? A review of recent evidence and theory. *Language Linguistics Compass*, e12434. <https://doi.org/10.1111/lnc3.12434>

===PREPARING YOUR MANUSCRIPT===

===PREPARING YOUR REVISION IN SCHOLARONE===

- Any electronic supplementary material (ESM).
- If you are requesting a discretionary waiver for the article processing charge, the waiver form must be included at this step.
- If you are providing image files for potential cover images, please upload these at this step, and inform the editorial office you have done so. You must hold the copyright to any image provided.
- A copy of your point-by-point response to referees and Editors. This will expedite the preparation of your proof.

- Ensure that your data access statement meets the requirements at <https://royalsociety.org/journals/authors/author-guidelines/#data>. You should ensure that you cite the dataset in your reference list. If you have deposited data etc in the Dryad repository, please include both the 'For publication' link and 'For review' link at this stage.
- If you are requesting an article processing charge waiver, you must select the relevant waiver option (if requesting a discretionary waiver, the form should have been uploaded at Step 3 'File upload' above).
- If you have uploaded ESM files, please ensure you follow the guidance at <https://royalsociety.org/journals/authors/author-guidelines/#supplementary-material> to include a suitable title and informative caption. An example of appropriate titling and captioning may be found at https://figshare.com/articles/Table_S2_from_Is_there_a_trade-off_between_peak_performance_and_performance_breadth_across_temperatures_for_aerobic_scope_in_teleost_fishes_/3843624.

Author's Response to Decision Letter for (RSOS-211082.R0)

See Appendix A.

RSOS-211082.R1 (Revision)

Review form: Reviewer 1

Is the manuscript scientifically sound in its present form?

Yes

Are the interpretations and conclusions justified by the results?

Yes

Is the language acceptable?

Yes

Do you have any ethical concerns with this paper?

No

Have you any concerns about statistical analyses in this paper?

No

Recommendation?

Accept as is

Comments to the Author(s)

I only had a couple of small suggestions. The authors did a very good job, and I recommend publication of this ms as it is now.

Review form: Reviewer 2 (Kevin Paterson)

Is the manuscript scientifically sound in its present form?

Yes

Are the interpretations and conclusions justified by the results?

Yes

Is the language acceptable?

Yes

Do you have any ethical concerns with this paper?

No

Have you any concerns about statistical analyses in this paper?

No

Recommendation?

Accept as is

Comments to the Author(s)

The authors have responded fully to my comments and I'm satisfied with their responses. I recommend acceptance of this manuscript.

Decision letter (RSOS-211082.R1)

Dear Dr Mirault,

It is a pleasure to accept your manuscript entitled "Fast priming of grammatical decisions: Repetition and transposed-word priming effects" in its current form for publication in Royal Society Open Science. The comments of the reviewer(s) who reviewed your manuscript are included at the foot of this letter.

on behalf of Professor Martin Pickering (Associate Editor) and Essi Viding (Subject Editor)
openscience@royalsociety.org

Reviewer comments to Author:

Reviewer: 1

Comments to the Author(s)

I only had a couple of small suggestions. The authors did a very good job, and I recommend publication of this ms as it is now.

Reviewer: 2

Comments to the Author(s)

The authors have responded fully to my comments and I'm satisfied with their responses. I recommend acceptance of this manuscript.

Appendix A

RSOS-211082

"Fast priming of grammatical decisions: Repetition and transposed-word priming effects"

Dear Professor Pickering,

Thank you for the invitation to submit a revision of this work, and many thanks to the reviewers for their constructive criticism. We have responded to all the points raised by the reviewer and we describe how we did so point-by-point below. We would also like to point out that several of the points raised by the reviewers were related to formatting problems and missing information that arose when converting our original Word doc into the RSOS template prior to submission, and we apologize for this. So, you will see that some of our responses just refer to these unfortunate errors / omissions.

All changes in the manuscript are indicated in yellow highlight in the highlighted version.

Hoping that you find these changes satisfactory.

Jonathan Mirault, Mathieu Declerck, Jonathan Grainger

Reviewer: 1

This is a very straightforward manuscript presenting a novel priming task together with an experiment using the task manipulation prime-target relationships. Clearly, the findings showed a priming effect, and this priming methodology can be extended to many different scenarios, so I am very positive.

I only have a minor comment. I believe the authors can make of their data by running some more exploratory analyses to shed some light on the nature of the priming effects—which were very large for repetitions but substantially smaller for transpositions. My suggestion is to obtain the delta plots for the repetition and transposition effects, as they may offer some extra interpretive cues (e.g., an early effect in the .1 quantile; whether the effect grows in the higher quantiles for repetition effects, but much less so for transposition effects, and so on). There are many examples of the use of delta plots in priming experiments (e.g., 10.1016/j.jecp.2020.104911, to cite one instance).

RESPONSE: Excellent suggestion. We now provide the requested delta plots and cite the suggested example. The delta plots show that our priming effects are quite robust across the entire distribution of RTs, except for the transposition effects with ungrammatical targets, which disappear in the longer RTs.

--Table 1 may be better formatted (there was an isolated “1” in the .pdf)

RESPONSE: This occurred in the transformation of the RSOS doc into pdf format. It should be ok now.

--What are the numbers in the parentheses in Table 2?

RESPONSE: These are 95% confidence intervals (CIs). This information was provided in the Captions for Figures and Tables at the end of the manuscript. We have now placed it as a note below the Table.

Reviewer: 2

This is a very well written paper that introduces a fast priming method to investigate elements of sentence-level processing for short sentences. The study is well-designed, with good use of pre-registration methods and a transparent account of where the methods deviated from this protocol. The authors also explicitly consider statistical power and take account of this in determining sample size.

Participants have to make grammatical judgments about short word strings. These strings follow the presentation of prime strings. In one case (repetition primes), the primes are structurally equivalent to the target and contain either the same words (the related condition) or different words (the unrelated condition). In other case (transposed-word primes), the primes include a word transposition but with either the same word (related condition) or different words (unrelated condition) to the target. The authors examine the effects of these primes on error rates and latencies for correct responses in the grammatical judgment task.

The key findings are that grammatical decisions are facilitated by the related relative to the unrelated primes in both the repetition priming and transposed-word conditions.

This is an interesting paradigm with some equally interesting results. However, there are several issues with the findings that I feel need further explanation. Also, the discussion of transposed-word effects would benefit from some discussion of alternative possibilities.

1. The key effect is that primes containing the same words facilitate grammatical decisions relative to primes with substituted words. This is particularly important in the context of the transposed-word primes as this shows that primes containing word transpositions work similarly to repetition primes, and can facilitate grammaticality decisions. This adds support to the authors claim that word order is processed flexibly in reading. I can see this is the case. However, the analyses they report do not compare the effect in repetition priming versus transposed-word priming. It isn't clear why this comparison is not made. Footnote 3 notes that this approach, which differs from the pre-registration, was because the related conditions for repetition and transposed word conditions are inherently different. I don't follow that argument, as the only difference appears to be the transposition of words to create the transposed-word primes.

Interestingly, Figure 2 appears to show longer responses for grammatical (and also ungrammatical) targets following a transposed-word prime than a repetition prime. There is also a much smaller related prime advantage for the transposed-word targets compared to the repetition targets. This seems to suggest a smaller benefit for transposed-word primes compared to repetition primes and therefore a cost for word transposition processing? How does this fit with arguments for flexible word order processing? Does it suggest this flexibility incurs a cost?

RESPONSE: We had already discussed this difference in priming effects for repetition primes and transposed-word primes in the penultimate paragraph of page 7. As is the case for letter order encoding (e.g., Rayner et al., 2006), flexibility in word order encoding does indeed come with a cost. Only an extreme version of word order flexibility would predict no cost, and we now point this out in new footnote 5.

2. I'm unclear why an ungrammatical prime was used for the ungrammatical targets. Also, I'm not clear why a transposed-word ungrammatical prime should facilitate decisions for these targets relative to the substitution condition. The explanation for transposed-word effects is that flexibility in the processing of word order allows access to a grammatical representation of the input that should then facilitate the subsequent recognition of a grammatical target. However, it's unclear how a transposed-word prime should facilitate grammaticality decisions for an ungrammatical target.

RESPONSE: The choice of ungrammatical primes for the ungrammatical targets was motivated by the fact that the repetition primes had to be ungrammatical, and therefore we matched the unrelated prime condition in terms of its grammaticality (i.e., both the repetition primes and the corresponding unrelated primes were ungrammatical). As concerns the priming effect we obtained for ungrammatical targets, we understand the confusion here, and have clarified all this in the revision. The important point to note is that the transposed-word primes for ungrammatical targets could not be resolved into a correct sentence by transposing two words (as was the case for the transposed-word primes for the grammatical targets). In other words, they would not have led to more evidence for a "grammatical" decision than an "ungrammatical" decision because they were generating more activity in a correct sentence representation. On the other hand, we would argue that in order to make an "ungrammatical" decision, you also need information about word identities and word order, and any of the related primes (i.e., repetition and transposed-word primes) in our study will boost activation in word identities and their associated positions (albeit much less so for transposed-word primes than repetition primes), hence the facilitation in making an "ungrammatical" decision following a related prime. This reasoning is now briefly mentioned in the first paragraph of the Discussion.

3. The argument presented here is that basic information about grammatical structure of at least a portion of a sentence can be extracted quickly during reading, from a 200ms display, and that this allows a coarse or "good enough" computation of grammatical structure. However, if basic information about grammatical structure can be obtained this quickly, why are responses in the grammatical judgment task so slow, in the order of 1300 to 1500 ms for grammatical targets in the present experiment?

RESPONSE: Good point. We had actually already addressed this issue in a different paper (Mirault et al., QJEP, 2020) where we show that relatively accurate grammatical decisions can be made with exposure durations as little as 300 ms. This was already mentioned in the Introduction. So the idea would be that the time that elapses between having enough information to make a reasonably accurate grammatical decision and the moment the actual decision is made (i.e., the button press) is taken up by decision-level and verification processes whereby an initial fast decision is subject to subsequent checking as more information becomes available with time and/or eye movements. We now mention this in new footnote 4.

4. I appreciate the authors have produced a streamlined report that is focused on the main issues motivating the use of the fast priming task to study sentence-level processing. However, in discussing word transposition effects, it would be helpful to acknowledge

controversies that sound this effect, including whether it relies on the parallel encoding of multiple words at a time during reading, as the authors claim, and whether the transposed word effect occurs perceptually or post-perceptually. I note, for example, the recent review article by Huang and Staub (2021).

Huang, K.-J., & Staub, A. (2021). Why do readers fail to notice word transpositions, omissions, and repetitions? A review of recent evidence and theory. *Language Linguistics Compass*, e12434. <https://doi.org/10.1111/lnc3.12434>

RESPONSE: We now acknowledge the current controversy with respect to the source of transposed-word effects and in particular whether or not they reflect parallel word processing, and in doing so we refer to the Huang and Staub paper.